# Data-Mining-Aided-Material Design of Doped LaMnO_3_ Perovskites with Higher Curie Temperature

**DOI:** 10.3390/ma18112437

**Published:** 2025-05-23

**Authors:** Lumin Tian, Wentan Wang, Xiaobo Ji, Zhibin Xu, Wenyan Zhou, Wencong Lu

**Affiliations:** 1College of Sciences, Shanghai University, Shanghai 200444, China; tianlumin39@163.com (L.T.); xiaoboji@t.shu.edu.cn (X.J.); 2College of Chemistry and Chemical Engineering, Beijing Institute of Technology, Beijing 100081, China; wwt182@126.com (W.W.); zbxu@bit.edu.cn (Z.X.); 3Sino-Platinum Metals Semiconductor Materials (Yunnan) Co., Ltd., Kunming 650101, China; 4Yunnan Precious Metals Laboratory Co., Ltd., Kunming 650106, China; 5State Key Laboratory of Advanced Technologies for Comprehensive Utilization of Platinum Metals, Kunming 650106, China

**Keywords:** Curie temperature, data mining, machine learning, perovskite, materials design

## Abstract

The Curie temperature (Tc) of LaMnO_3_-based perovskites is one of the most important properties associated with their magnetic and spintronic applications. The search for new perovskites with even higher Tc is a challenging problem in material design. Through the systematic optimization of support vector regression (SVR) architecture, we establish a predictive framework for determining the Curie temperature (Tc) of doped LaMnO_3_ perovskites, leveraging fundamental atomic descriptors. The correlation coefficient (R) between the predicted and experimental Curie temperatures demonstrated high values of 0.9111 when evaluated through the leave-one-out cross-validation (LOOCV) approach, while maintaining a robust correlation of 0.8385 on the independent test set. The subsequent high-throughput screening of perovskite compounds exhibiting higher Curie temperatures was implemented via our online computation platform for materials data mining (OCPMDM), enabling the rapid identification of candidate materials through systematic screening protocols. The findings demonstrate that machine learning exhibits significant efficacy and cost-effectiveness in identifying lanthanum manganite perovskites with elevated Tc, as validated through comparative computational and empirical analyses. Furthermore, a web-based computational infrastructure is implemented for the global dissemination of the predictive framework, enabling the open-access deployment of the validated machine learning model.

## 1. Introduction

Perovskite exhibits broad applicability in fields such as hydrogen generation via photocatalytic water splitting and photovoltaic devices, attributed to its exceptional ferroelectric behavior, versatile physicochemical functionalities, and facile structural tailorability [1,2]. Nevertheless, the exploration of perovskite-based functionalities is largely limited by the insufficiently high Curie temperatures (Tc) of existing materials. Tc is the temperature at which a system undergoes a phase transition from ferromagnetic to paramagnetic, or the temperature at which transition from ferroelectric to paraelectric ordering takes place. Enhancing the Curie temperature of perovskite materials remains a critical challenge requiring further investigation in this research field [3,4,5].

With the constant advancement of the Materials Genome Initiative (MGI), an increasing number of data mining models have been reported to facilitate the acceleration of material design and optimization. An increasing number of researchers are contributing to the rational design and property customization of advanced materials through the strategic manipulation of material “genes”. These genes are atomic-scale descriptors that include composition, structural characteristics, defect states, and synthesis parameters, all of which influence macroscopic functionalities. Data-driven material design also establishes a multidisciplinary framework that synergistically couples computational datasets with experimental validation through the integration of high-throughput computational screening and multiscale simulation techniques. This paradigm transition replaces conventional empirical trial-and-error approaches with a rational design strategy, enabling cost-effectiveness, efficiency, and accelerated development timelines. The systematic convergence of these methodologies facilitates the rapid exploration of material property landscapes, thereby substantially lowering research and development expenditures and the time-to-market for advanced functional materials. Sun et al. developed a data-driven surrogate model to predict the solar array output power in stratospheric airships, which achieved a remarkable improvement in prediction accuracy by 98.65% compared to conventional numerical simulations, while demonstrating a 10-million-fold increase in computational efficiency [6]. Li et al. reported an enhanced performance in inverted perovskite solar cells through chemical interface engineering, achieving an impressive power conversion efficiency (PCE) of 25% [7].

As the typical colossal magnetoresistive perovskites, the search for new lanthanum manganite (LaMnO_3_)-based perovskites with a higher Tc is a challenging problem for emerging spintronics. Herein, we develop a machine learning model to screen doped LaMnO_3_ perovskites with a high temperature performance. The structural configuration of the ABO_3_-type perovskites examined in this study is schematically represented in Figure 1, demonstrating the characteristic cation occupancy where the A-site is predominantly occupied by rare-earth/alkaline-earth species (e.g., La^3+^, Sr^2+^), and the B-site by transition metal cations (e.g., Mn^3+^). The A and B lattice positions in the A-type antiferromagnetic [8] insulator LaMnO_3_ permit cation substitution to modulate its functional properties. For instance, the substitution of some La^3+^ by divalent cations like Sr^2+^ or Ca^2+^ induces significant electronic effects, which form the basis of enormous complex phenomena, including colossal magnetoresistance.

It should be noted that many theoretical and experimental researchers might criticize machine learning models, because these algorithms are often utilized as a *black box* in most case studies. In this work, we will not discuss the validity of the criticism and possible approaches to this challenge. Nevertheless, we have provided our developed models by allowing anyone to use the tool online easily, as it is accessible via a public website. Thus, we hope that it will guide future experiments to accelerate the search for lanthanum manganite perovskites with a higher Tc.

## 2. Methods

### 2.1. The Flowchart of Materials Data Mining

Figure 2 illustrates the six steps of material data mining undertaken in this study. To develop a robust machine learning model for assisting the design of lanthanum manganite perovskite materials with an elevated critical temperature (Tc), the following six-step protocol should be implemented: (i) *Data preparation* involves the compilation of a rigorously validated benchmark dataset to facilitate the training and testing of computational models; (ii) *Feature selection* requires us to choose the optimal subset from the initial features as the descriptors of the model; (iii) *Model selection* demonstrates the establishment of a statistical optimization framework that systematically captures the underlying correlations between the response variables and the prediction space; (iv) *Hyper-parameter optimization* suggests how to adjust the parameters adopted in a model so that the model can perform well; (v) *Model Validation* refers to the process of assessing the predictive accuracy of a developed model through cross-validation tests and independent tests; (vi) *Model Application* establishes a workflow integrating predictive modeling with material design, culminating in the deployment of an automated web platform that enables the accelerated discovery of candidate materials through cloud-based computational screening. Below, we detail the contents of the six steps mentioned above.

### 2.2. Data Preparation

We collected 66 example compounds of La_x_M_y_N_z_R_1-x-y-z_Mn_n_Q_1-n_O_3_ perovskites (M, N, R, and Q are dopant metal cations) with a Tc ranging between 160 K and 380 K from the literature (see Appendix A) [9,10,11,12,13,14,15,16,17,18,19,20,21,22,23,24]. The dataset was divided into two subsets in a random way: 53 samples for the training and the leave-one-out cross-validation (LOOCV) set, and 13 samples for the test set. In addition, we collected 21 atomic parameters as the initial features quoted from Lange’s Handbook of Chemistry (16th ed.) [25], as tabulated in Table 1.

### 2.3. Machine Learning Methods

#### 2.3.1. Gradient Boosting Regression (GBR)

Gradient boosting is an integrated learning algorithm that is based on the idea of adding new trees to the ensemble in order [26]. It can avoid the problem of overfitting with good accuracy. At each iteration, GBR minimizes the loss function by recovering the error of the previous ensemble trees while performing prediction in the next tree [27]. Thus, the error will keep decreasing. GBR has several advantages, such as excellent adaptability to high-dimensional mixed types of input values, including numerical and categorical variables and robustness against irrelevant input variables, which means that we do not need to input the missing values because the model can handle it automatically. In addition, the model does not need to be retrained from the very beginning when new data are available, therefore improving the calculation efficiency greatly. Because of these multiple advantages, GBR has been widely used in many research fields [28,29,30].

#### 2.3.2. Decision Tree Regression (DTR)

Decision tree is a data analytic technique that explores the potential complex interactions within data by creating binary segmentations of individuals into sub-groups [31]. The membership of a subgroup is derived from the response to a set of measurement/prediction variables. Individuals are initially grouped based on similar scores for one predictor, and then divided into different subgroups based on further predictors [32]. And the position of the variables is determined hierarchically by locating the most important variables at the root of the tree [33]. Decision tree has many advantages. For example, it does not need to assume the distribution of exploratory variables; it is not affected by the high correlation between independent variables; and the most important variables that explain the dependent variables are included in the decision tree, while the unimportant variables are excluded [34]. There are various decision tree methods, including chi-squared automatic interaction detection (CHAID), the classification and regression tree (CART), and exclusive CHAID. Although decision tree was originally developed for large datasets, it may also provide accurate predictions for small datasets.

#### 2.3.3. Random Forest Regression (RFR)

The random forest algorithm, initially proposed by Breiman and rooted in decision tree architectures, exhibits enhanced generalization capabilities as an ensemble learning methodology [35]. RFR constitutes an ensemble of unpruned regression trees constructed through the bootstrap sampling of training data with random feature selection during tree construction, ultimately aggregating predictions through mean averaging across all constituent trees in the regression framework [36]. In other words, random forest aims to find a consensus among the imperfect trees rather than a perfect one which may result in overfitting. This algorithm is well suited to dealing with high-dimensional datasets, avoiding overfitting and allowing an averaging effect across all the single models [37,38].

#### 2.3.4. Support Vector Regression (SVR)

Support vector machine (SVM) was initially introduced by Vapnik in 1964, and is grounded in the framework of statistical learning theory (SLT) [39]. This supervised learning algorithm employs risk minimization principles by strategically balancing empirical risk and expected risk, thereby deriving globally optimal solutions through the convex optimization of the regularized loss function. SVM, encompassing support vector classification (SVC) and support vector regression (SVR), has been widely utilized to address nonlinear classification and regression challenges owing to its demonstrated robustness in handling small-scale datasets characterized by high-dimensional feature spaces [40,41,42].

Given the training data {(x_i_, y_i_), i = 1, 2, … l}, x ∈ R_n_, y ∈ R, the regression function is defined as the following linear relationship:(1)$$f\left(x\right)=w^{T}x+b$$
where *w* and *b* are the coefficients to be adjusted. The optimal regression function can be obtained by calculating the minimum value of the following formula:(2)$$\varphi \left(w,\xi^{*},\xi \right)=\frac{1}{2}{||w||}^{2}+C\left(\overset{n}{\underset{i=1}{\sum }}\xi_{i}+\overset{n}{\underset{i=1}{\sum }}\xi_{i}^{*}\right)$$
where *C* is the regularization constant, which adjusts the trade-off between the model complexity and training errors. ξ and $\xi^{*}$ are the upper and lower limits of slack variables. Vapnik put forward the ɛ-insensitive loss function:(3)$$L_{e}\left(y\right)=\left\{\begin{array}{c} \;0\;\left|f\left(x\right)-\left(y\right)\right|<\varepsilon \\ \left|f\left(x\right)-\left(y\right)\right|-\varepsilon \;otherwise \end{array}\right.$$

This function aims to determine an optimal hyperplane that maximizes the margin of separation between two linearly separable subsets within the training dataset. The solution can be obtained by the following quadratic programming optimization problem shown in Equation (4)(4)$$\underset{\alpha ,\alpha^{*}}{\max}W\left(\alpha ,\alpha^{*}\right)=\underset{\alpha ,\alpha^{*}}{\max}-\frac{1}{2}\overset{l}{\underset{i=1}{\sum }}\overset{l}{\underset{j=1}{\sum }}\left(\alpha_{i}-\alpha_{i}^{*}\right)\left(\alpha_{j}-\alpha_{j}^{*}\right)x^{T}x_{j}+\overset{l}{\underset{i=1}{\sum }}\alpha_{i}\left(y_{i}-\varepsilon \right)-\alpha_{i}^{*}\left(y_{i}+\varepsilon \right)$$

The constraints of this optimization problem are as follows:(5)$$\left\{\begin{array}{c} \overset{l}{\underset{i=1}{\sum }}\left(\alpha_{i}^{*}-\alpha_{i}\right)=0\\ 0\leq \alpha_{i},\alpha_{i}^{*}\leq C\;i=1,\cdot \cdot \cdot ,l \end{array}\right.$$
where $\alpha_{i}\;$and $\alpha_{i}^{*}\;$are the Lagrangian coefficients. By solving the above-described optimization problem, the coefficients of Equation (1) can be found as follows:(6)$$\bar{w}=\overset{l}{\underset{i=1}{\sum }}\left(\alpha_{i}-\alpha_{i}^{*}\right)x_{i}$$

For the nonlinear relationship between the input and output, the kernel function (K) is introduced to map the original input into the feature space nonlinearly; thus, SVR can be used for more complicated nonlinear regression problems. Within the feature space, the kernel function is formally expressed as follows:(7)$$K\left(x_{i},x_{j}\right)=<\Phi \left(x_{i}\right)\cdot \Phi \left(x_{j}\right)>$$

The predominantly employed kernel functions encompass the Gaussian radial basis function (RBF) and polynomial kernels. In this study, the RBF is utilized in the SVR model as follows:(8)$$K\left(x_{i},x_{j}\right)=\exp\left(\frac{-{||x_{i}-x_{j}||}^{2}}{\sigma^{2}}\right)$$

### 2.4. Computational Software

The material data mining was carried out by using the ExpMiner software package (http://materials-data-mining.com/home/static/download/ExpMiner_setup.msi (15 May 2025)) and the Online Computational Platform of Material Data Mining (OCPMDM) developed in our laboratory [43,44]. ExpMiner can be freely downloaded from the website of the Laboratory of Materials Data Mining in Shanghai University (http://materials-data-mining.com/home/static/download/ExpMiner_setup.msi (accessed on 15 May 2025)), and the OCPMDM can be accessed online at http://materials-data-mining.com/ocpmdm/ (accessed on 15 May 2025).

## 3. Results and Discussion

### 3.1. Feature Selection

For any machine learning model, feature selection is very critical because it is difficult to eliminate unnecessary features to improve the prediction performance. The success of a model is determined by the key features selected from a great deal of candidate features. Proper feature selection can not only reduce the dimension of the feature space without redundancy, but also shorten the training time, and further improve the precision and performance of the model [45].

In this work, GA (Genetic Algorithm) [46] combined with SVR, FR (Forward Regression) and BR (Backward Regression) was adopted to select the key features, respectively. The GA approach simulates the natural selection of Darwin’s biological evolution and the biological evolution process of the genetic mechanism. It is a kind of randomization search method derived from the evolution law of biology (survival of the fittest). Compared with other algorithms, GA can move from the local optima present on the response surface and realize a wide variety of optimizations without requiring knowledge or a gradient about the response surface to be present. FR and BR are two classical methods used for independent variable screening, whose principles are simple but effective. Forward selection is performed to keep one variable at the beginning, and then other variables are added step by step [47]. Meanwhile, the contribution of variables to the model is observed, keeping the variables with a significant contribution and eliminating the ones with a small contribution until the model reaches an optimal level. On the contrary, the backward selection is performed to adopt all the variables at the beginning, and then gradually eliminate the variables that have no significant contribution to the model until all the variables in the model have a significant contribution.

In order to evaluate the feature selection, the root mean square error (RMSE) was employed as the goodness-of-fit measure. The RMSE is defined as follows:(9)$$\mathrm{RMSE}=\sqrt{\frac{\sum_{i=1}^{n}{\left(p_{i}-e_{i}\right)}^{2}}{n}}$$
where $e_{i}$ and $p_{i}$ are the experimental and predicted values of sample i, respectively, and n is the number of samples. A statistically significant inverse relationship is observed between RMSE magnitudes and the optimality of the feature set, where diminished RMSE values systematically correspond to an enhanced predictive performance in computational modeling frameworks.

A total of 21 initial features (all are listed in Appendix A) such as the tolerance factor, atomic radius and molecular mass were chosen to describe the perovskites. To mitigate the redundancy and interdependence among the 21 features, their Pearson correlation coefficients were computed and are presented in Figure 3a. After checking the correlation coefficients between pairs of features, it was found that four features could be deleted because their correlation coefficients with the other features were more than 0.9. As shown in Figure 3b, the decorrelation process resulted in a reduction in the selected feature set from 21 to 17.

Then, GA combined with SVR, BR and FR was employed to filter the features. According to the LOOCV results, FR outperformed the other two algorithms, as shown in Table 2. Thus, the seven features from the forward method based on the LOOCV results obtained by using SVR were selected to establish the model as the optimal subset of initial features.

### 3.2. Model Selection

The development of an optimal model that achieves a balance between overfitting and underfitting is critical in data mining. Here, the operational validation of different machine learning models was executed through LOOCV procedures, employing correlation coefficient R as the key performance indicator for predictive consistency measurement. In this work, four algorithms, namely, SVR, GBR, RFR and DTR, were used to predict the Tc of perovskites, whose results are shown in Table 3 and Figure 4. Comparative analyses showed that SVR outperformed the other models based on the R, RMSE and MRE (mean relative error) of LOOCV tests. Thus, the SVR algorithm was selected to construct the prediction model. The MRE is defined as follows:(10)$$\mathrm{MRE}=\frac{100\%}{n}\overset{n}{\underset{i=1}{\sum }}\left|\frac{e_{i}-p_{i}}{e_{i}}\right|$$
where $e_{i}$ and $p_{i}$ denote the experimental and predicted values of sample i, respectively, and n represents the total sample count.

### 3.3. Hyper-Parameter Optimization

The model selection identified Support Vector Regression (SVR) as the optimal candidate, demonstrating statistical superiority, with the highest R = 0.8849 and lowest RMSE (26.50) compared to other algorithms. To improve its generalization ability, the SVR model using the Gaussian radial basis function (RBF) was further optimized based on the hyper-parameter optimization. The optimal model performance was achieved with parameter values of C = 32, ε =0.06, and γ =1.0, as illustrated in Figure 5.

### 3.4. Model Validation

To validate the generalization capability of the SVR model, LOOCV was employed to assess the predictive robustness of the developed framework, thereby ensuring the rigorous evaluation of its extrapolation performance. Figure 6 shows the plots of the predicted values versus the experimental values of Tc for lanthanum manganite perovskites based on the LOOCV of the training set. It is observed that the predicted values lie near the actual line (y = x), with R and RMSE values equal to 0.9119 and 23.22 (K), respectively, as shown in Table 4. The MAE is defined as follows:(11)$$\mathrm{MAE}=\frac{1}{n}\overset{n}{\underset{i=1}{\sum }}\left|p_{i}-e_{i}\right|$$
where $e_{i}$ and $p_{i}$ are the experimental and predicted values of sample i, respectively, and n is the number of samples.

To further confirm the generalization performance of the SVR model, the Curie temperatures of the 13 samples in the independent test set were predicted by using the trained model. Figure 7 illustrates the experimental and predicted Tc values of the training and test datasets, respectively. The R of the independent test is 0.8385, which is shown in Table 5.

From the above results, it can be concluded that the SVR model established in this work would be able to predict the Tc of lanthanum manganite perovskites.

### 3.5. Sensitivity Analysis

Sensitivity analysis has been extensively applied across various data mining domains to investigate the variation in the target variable with respect to an individual feature, while maintaining other features constant at sensitivity points. The SVR model available here shows that some parameters are significant with regard to the Tc of perovskites. Figure 8 delineates the sensitivity assessment for each feature in the optimized subset.

Figure 8a,b show that the Tc goes up with the increase in R_a/R_b and A_aff, respectively. Figure 8c–e,g demonstrate that B_aff, B_Tm, A_Tb and A_Hfus have parabolic relationships with Tc, with a maximum Tc equal to 392.7, 345.7, 340.5 and 342.6, respectively. In Figure 8f, B_Tb is negatively correlated with Tc.

### 3.6. Model Application

#### 3.6.1. High-Throughput Screening of New Lanthanum Manganite Perovskites

In order to discover new perovskites with a higher Tc, the established model for predicting Tc was used on OCPMDM to screen out the targeted perovskites among various candidates. The screening was performed according to the following rules:(1)The A site is doped with no more than two different doping ions, while the B site is doped with no more than one doping ion.(2)The A site compositional configuration follows a ternary doping scheme:Primary occupant: La with stoichiometric ratios spanning 0.5–1.0 in 0.02 increments.Secondary dopant: Sr, Ag, Ca, Pb, or Ba allocated within the 0.0–0.5 range (0.02 step resolution).Tertiary constituent: Nd, Ba, Ag, Ca, or Dy occupying residual stoichiometric fractions.
(3)The B-site doping architecture adopts a two-component system:Primary constituent: Mn with stoichiometric fractions ranging 0.9–1.0 in 0.02 incremental steps.Secondary dopant: Fe or Cr occupying the complementary stoichiometric proportion.

Thus, in total, 99,900 candidates were obtained fir screening. It was found that the highest Tc (391.7 K) of the candidate sample was La_0.54_Ag_0.14_Ba_0.32_MnO_3_, which has a tolerance factor of 0.89 and exceeds the highest Tc in the training data set (375 K). We hope that the perovskite with a predicted high Tc (391.7 K) will be experimentally verified soon in the laboratory.

#### 3.6.2. Online Web Server Accessible in Public

To facilitate the application of the developed SVR model in perovskite design, a web-based computational platform was implemented to predict the Tc of lanthanum manganite perovskites, leveraging the predictive framework established in this study. Figure 9 shows the interface of the online server for utilizing our model. Once the chemical formula of perovskite is input and the *Predict* button is pressed, the Curie temperature predicted by the model available in this paper can be obtained. In addition, users can also add the model description, model creator and paper DOI for the work. The online server is accessible via the following URL: http://materials-data-mining.com/ocpmdm/material_api/negr5q1xunlmjagj (accessed on 15 May 2025).

## 4. Conclusions

Based on the data collected from the published references, we developed an SVR machine learning model to predict the Curie temperature of lanthanum manganite perovskites by using atomic parameters as inputs. The machine learning model showed a good performance when predicting Tc in a fast and easy way. Based on the high-throughput screening of visual candidates, we discovered the specific perovskite compound (La_0.54_Ag_0.14_Ba_0.32_MnO_3_) with a higher Curie temperature than the highest one in the training set. It is thought that other properties of materials could be optimized using the reliable data collected and the QSPR model constructed at a low cost [48,49]. As demonstrated in various publications [50,51,52,53,54], the Curie temperature of lanthanum manganite perovskites is affected by many factors such as the types and proportions of doping ions, their morphology, the experimental conditions and so on, which may lead to the deviation of the model established in this paper. Consequently, subsequent research phases will prioritize methodological refinement, including the reporting of more samples and the factors available for improving the prediction accuracy of the model in material design.

## Figures and Tables

**Figure 1 materials-18-02437-f001:**
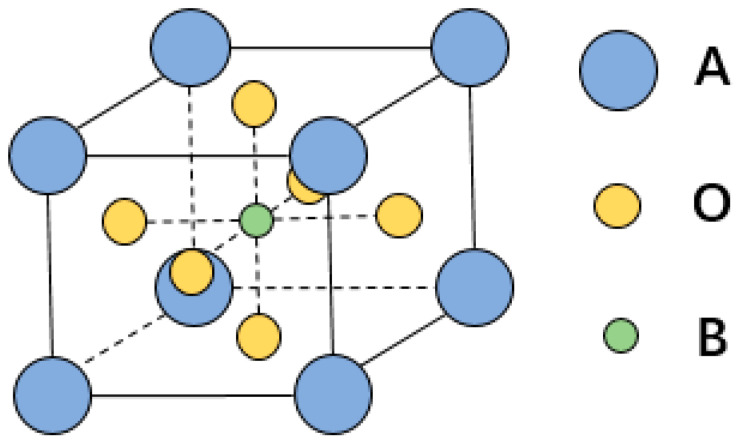
The crystal structure of ABO_3_ perovskite where the A-site cation is usually a rare-earth/alkaline-earth element like La^3+^ and Sr^2+^, while the B-site cation is a transition metal element such as Mn^3+^.

**Figure 2 materials-18-02437-f002:**
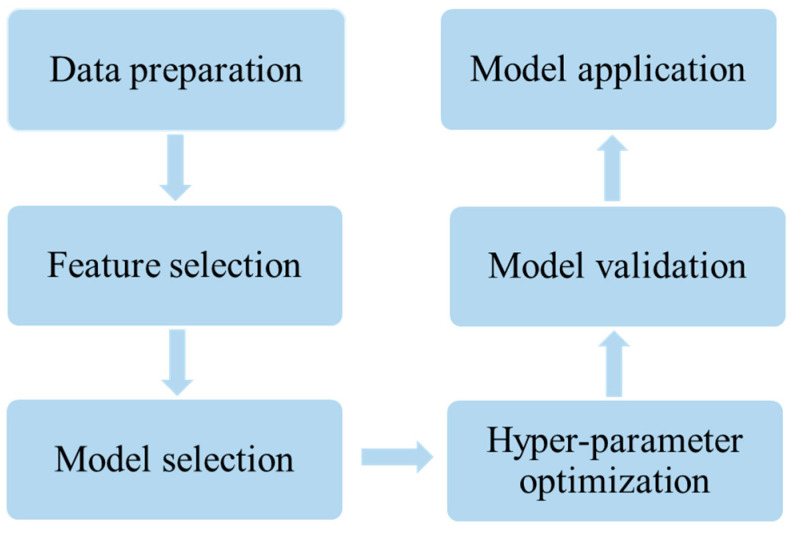
The flowchart of the six steps undertaken during material data mining in this work.

**Figure 3 materials-18-02437-f003:**
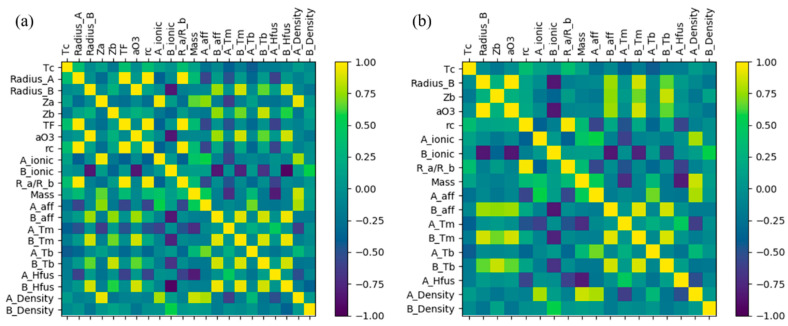
Pearson correlation coefficient heat map of (**a**) 21 initial features and (**b**) 17 selected features after deleting the features with co-linearity. The color-coded bar adjacent to the figure represents correlation coefficients, with yellow denoting positive correlations and blue signifying negative correlations.

**Figure 4 materials-18-02437-f004:**
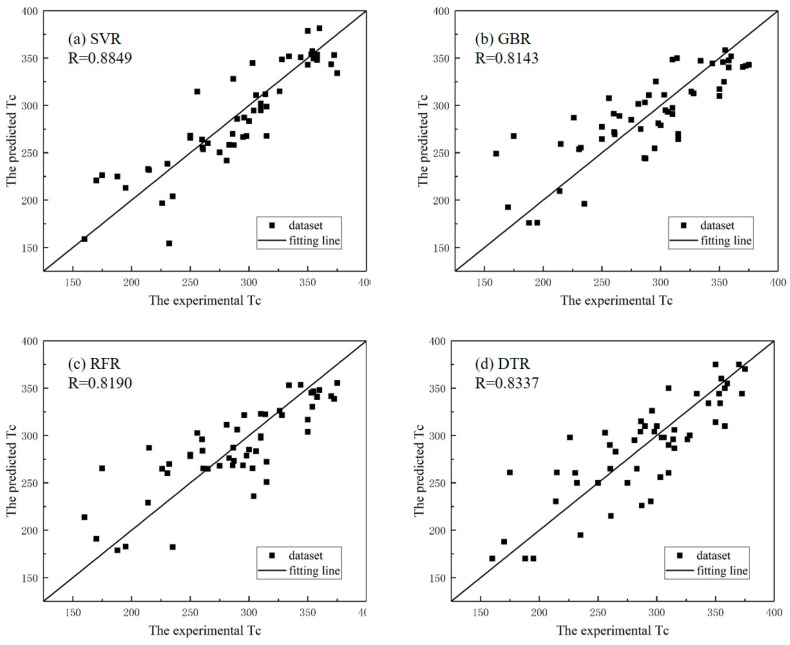
The experimental versus the predicted Tc of the LOOCV results when using (**a**) SVR, (**b**) GBR, (**c**) RFR and (**d**) DTR models.

**Figure 5 materials-18-02437-f005:**
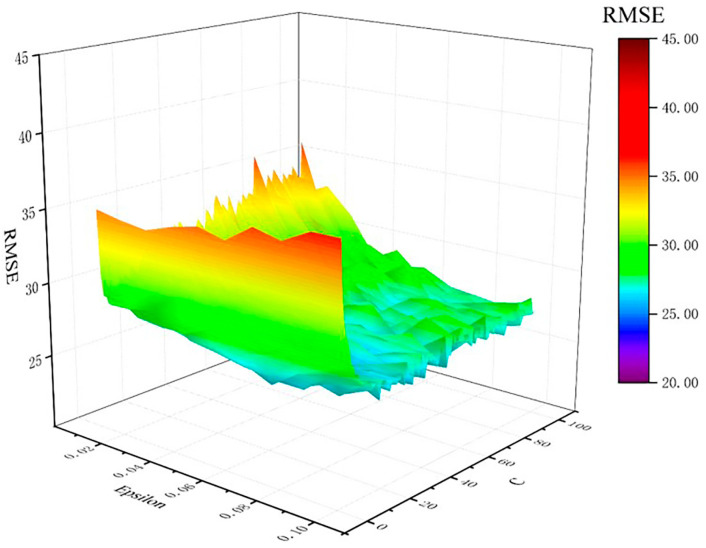
RMSE of LOOCV vs. ε and C adopted in the SVR model.

**Figure 6 materials-18-02437-f006:**
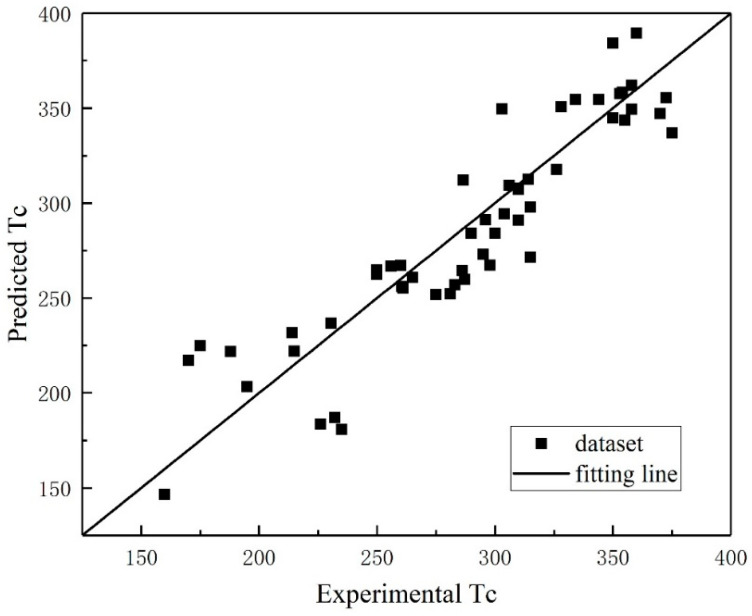
The experimental versus the predicted Tc based on the LOOCV results.

**Figure 7 materials-18-02437-f007:**
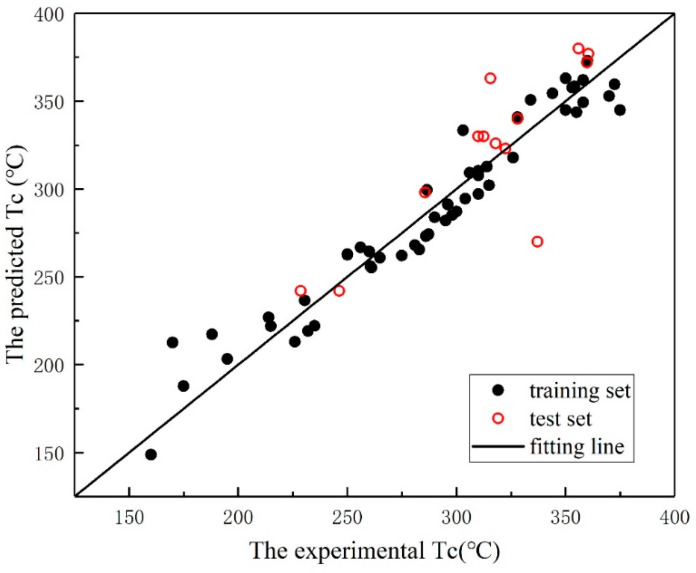
The experimental versus the predicted Tc of perovskites when using SVR.

**Figure 8 materials-18-02437-f008:**
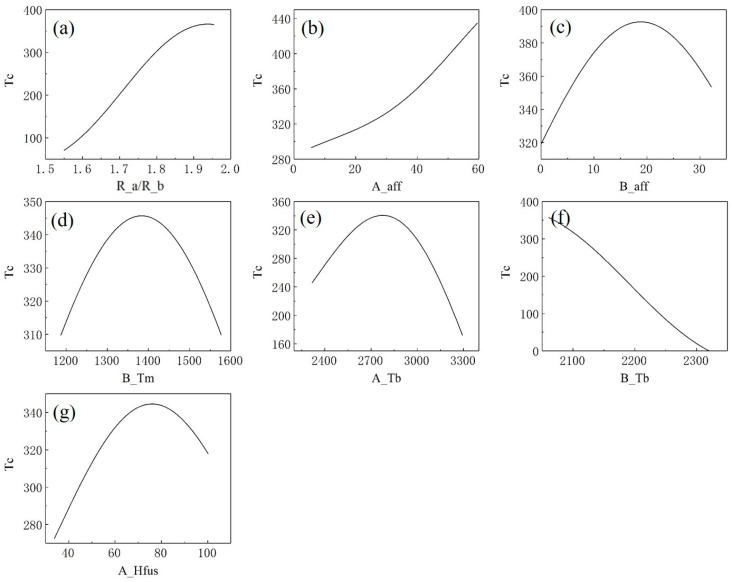
Sensitivity analyses with (**a**) the ratio of the atomic radius of the A and B positions (R_a/R_b), (**b**) the electron affinity of the A position (A_aff), (**c**) the electron affinity of the B position (B_aff), (**d**) the melting point of the B position (B_Tm), (**e**) the normal boiling point of the A position (A_Tb), (**f**) the normal boiling point of the B position (B_Tb), (**g**) the enthalpy of fusion at the melting point of the A position (A_Hfus). The sensitivity point is set at the average values of features of samples with a higher Tc.

**Figure 9 materials-18-02437-f009:**
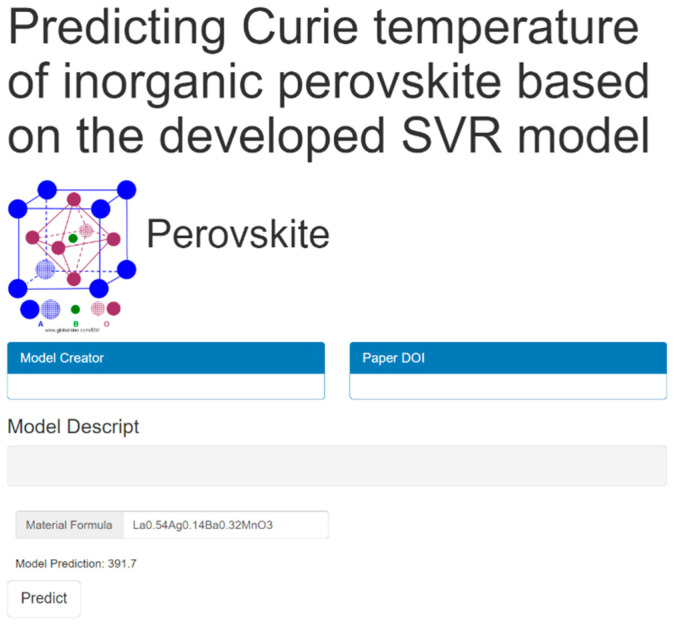
The interface of the online web server accessible to the public.

**Table 1 materials-18-02437-t001:** The 21 atomic parameters used as the initial features of the dataset.

No.	Feature Name	Feature Description
01	Radius_A	atomic radius of the A position
02	Radius_B	atomic radius of the B position
03	Ea	electronegativity of the A position
04	Eb	electronegativity of the B position
05	TF	tolerance factor
06	αO3	unit cell lattice edge
07	rc	critical radius
08	Za	ionization potential of the A position
09	Zb	ionization potential of the B position
10	R_a/R_b	ratio of the atomic radius of the A and B positions
11	Mass	molecular mass
12	A_aff	electron affinity of the A position
13	B_aff	electron affinity of the B position
14	A_Tm	melting point of the A position
15	B_Tm	melting point of the B position
16	A_Tb	normal boiling point of the A position
17	B_Tb	normal boiling point of the B position
18	A_Hfus	enthalpy of fusion at the melting point of the A position
19	B_Hfus	enthalpy of fusion at the melting point of the B position
20	D_A	density of the A position
21	D_B	density of the B position

**Table 2 materials-18-02437-t002:** The results of feature selection when using different algorithms.

Algorithm	RMSE (K)	R	The Number of Selected Features	The Selected Features
Genetic Algorithm	38.72	0.7658	7	Zb, aO3, A_ionic, R_a/R_b, A_aff, B_aff, A_Tb
Forward Regression	26.51	0.8849	7	R_a/R_b, A_aff, B_aff, B_Tm, A_Tb, B_Tb, A_Hfus
Backward Regression	28.29	0.8632	4	R_a/R_b, A_aff, A_Tb, B_Tb

**Table 3 materials-18-02437-t003:** The results of the LOOCV validations when using different algorithms, including SVR, GBR, RFR and DTR.

Algorithm	R	RMSE (K)	MRE
SVR	0.8849	26.50	0.0790
GBR	0.8143	32.19	0.1011
RFR	0.8190	31.82	0.0972
DTR	0.8337	31.29	0.0937

**Table 4 materials-18-02437-t004:** The LOOCV validation results of the training dataset.

Algorithm	R	RMSE (K)	MAE (K)	MRE
SVR	0.9111	23.82	18.94	0.0720

**Table 5 materials-18-02437-t005:** The results of the independent test.

Algorithm	R	RMSE (K)	MAE (K)	MRE
SVR	0.8385	26.36	19.67	0.0623

## Data Availability

The data presented in this study are openly available in the Appendix A.

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
