# Peer review of "Data-Mining-Aided-Material Design of Doped LaMnO3 Perovskites with Higher Curie Temperature"

_materials, 2025, doi:10.3390/ma18112437_

Round 1

Reviewer 1 Report

Comments and Suggestions for Authors

The authors present a tool to predict the Curie temperature of the doped LaMnO3 perovskite system using machine learning algorithms and several atomic parameters. The article is well presented and accurately describes how the authors reached the online tool. A few observations can improve the paper.

How was the ratio between the training/test data calculated?

What algorithm, if any, was used to ensure the randomness of the dataset test

Please state which 4 of the 21 parameters were omitted because of the correlation.

Reviewer 2 Report

Comments and Suggestions for Authors

Thank you for submitting your manuscript entitled "Data Mining Aided Materials Design of Doped-LaMnO₃ Perovskites with Higher Curie Temperature." This work addresses an important topic in computational materials discovery and magnetic materials. The combination of machine learning (ML) and high-throughput screening is highly relevant and appreciated.

However, the paper requires major revisions before it can be recommended for publication. Below are my specific comments and suggestions for improving the manuscript.

1. Introduction: Needs Broader Context and Key Citations
The current introduction is quite narrow in scope and does not sufficiently emphasize the broader scientific context or practical importance of high-Tc perovskite materials. I recommend a significant revision of the introduction section to include recent studies and extended applications of perovskite and data-driven materials design.

Please consider incorporating and discussing the following relevant and recent references, which will enrich the background and demonstrate awareness of the state-of-the-art:

Sun et al. (2024) – Developed a surrogate model for solar array output under global wind conditions, demonstrating the role of data science in performance prediction.
Chinese Journal of Aeronautics.
https://doi.org/10.1016/j.cja.2024.09.020

Li et al. (2024) – Reported improved performance in inverted perovskite solar cells through chemical interface engineering. This provides another angle on how compositional tuning enhances device function.
Advanced Energy Materials, 2404335.
https://doi.org/10.1002/aenm.202404335

Hao et al. (2024) – Presented work on full-spectrum solar absorption using composite structures, relevant for light-heat conversion and functional design in perovskite-like materials.
Chemical Engineering Journal, 497, 154979.
https://doi.org/10.1016/j.cej.2024.154979

These papers will help anchor your study in the broader efforts of materials design, thermal optimization, and perovskite functionalization — all of which are thematically close to your manuscript. Please refer to them in context when discussing recent strategies in high-Tc materials and computational screening.

2. Model Methodology: Needs Expansion and Detail
The SVR model is mentioned, but important methodological details are missing. Please provide:

Full feature list used as input (e.g., electronegativity, ionic radius?);

Normalization/scaling method;

Hyperparameter optimization approach;

Size of training/test datasets and split criteria;

Description of cross-validation strategy (is LOOCV used on full data or only part?).

Without this information, the model is not reproducible nor scientifically verifiable.

3. Results: Clarification and Visualizations Needed
Add visual summaries such as:

Residual error distributions;

A table listing top-ranked candidate perovskites with predicted Tc and composition.

Explain how the “OCPMDM” platform was built and whether it is available for the public or reproducible for independent validation.

4. English Language and Scientific Writing
Several grammatical and stylistic issues exist throughout the manuscript. Examples:

"The search of new perovskites" → should be "search for new perovskites"

"trail-and-error" → should be "trial-and-error"

"The Tc is one of the most important properties associated..." → consider rewording to: "The Curie temperature (Tc) is a key parameter that governs the magnetic performance and practical applications of LaMnO₃-based perovskites."

A thorough professional language edit is recommended.

Final Recommendation:
Please revise the manuscript thoroughly and address the above points. Once the scientific details are clarified, the model is made more reproducible, and the introduction is enhanced with the suggested references, the paper may be reconsidered.

Comments on the Quality of English Language

The English in the manuscript needs significant revision. Several grammatical mistakes, awkward phrasing, and inconsistencies are present. The authors should seek the assistance of a native speaker or professional editing service to improve clarity, readability, and technical expression.

Round 2

Reviewer 2 Report

Comments and Suggestions for Authors

ACCEPT